# Temporal Stability of Dynamic Default Mode Network Connectivity Negatively Correlates with Suicidality in Major Depressive Disorder

**DOI:** 10.3390/brainsci12091263

**Published:** 2022-09-17

**Authors:** Xuan Ouyang, Yicheng Long, Zhipeng Wu, Dayi Liu, Zhening Liu, Xiaojun Huang

**Affiliations:** 1Department of Psychiatry, National Clinical Research Center for Mental Disorders, The Second Xiangya Hospital, Central South University, Changsha 410011, China; 2Department of Psychiatry, Jiangxi Provincial People’s Hospital, The First Affiliated Hospital of Nanchang Medical College, Nanchang 330006, China

**Keywords:** major depressive disorder, suicide, neuroimaging, fMRI, dynamic functional connectivity, dynamic brain network

## Abstract

Previous studies have demonstrated that the suicidality in patients with major depressive disorder (MDD) is related to abnormal brain functional connectivity (FC) patterns. However, little is known about its relationship with dynamic functional connectivity (dFC) based on the assumption that brain FCs fluctuate over time. Temporal stabilities of dFCs within the whole brain and nine key networks were compared between 52 MDD patients and 21 age, sex-matched healthy controls (HCs) using resting-state functional magnetic resonance imaging and temporal correlation coefficients. The alterations in MDD were further correlated with the scores of suicidality item in the Hamilton Rating Scale for Depression (HAMD). Compared with HCs, the MDD patients showed a decreased temporal stability of dFC as indicated by a significantly decreased temporal correlation coefficient at the global level, as well as within the default mode network (DMN) and subcortical network. In addition, temporal correlation coefficients of the DMN were found to be significantly negatively correlated with the HAMD suicidality item scores in MDD patients. These results suggest that MDD may be characterized by excessive temporal fluctuations of dFCs within the DMN and subcortical network, and that decreased stability of DMN connectivity may be particularly associated with the suicidality in MDD.

## 1. Introduction

Major depressive disorder (MDD) is one of the leading causes of disability worldwide, partly because of the high rates of suicide attempts in MDD patients [1,2]. Neuroimaging studies using the resting-state functional magnetic resonance imaging (fMRI) have documented that the changes in functional connectivity (FC) between specific brain regions, such as increased precuneus-motor connectivity [3], increased amygdala-precuneus connectivity [4] and decreased fronto-limbic connectivity [5], are associated with increased suicidality in MDD. These suicidality-related alterations in brain functions may have important implications for understanding and preventing suicide among MDD patients.

The conventional fMRI studies assume that the patterns of brain FC are stationary. Recently, the brain FC was found to fluctuate over time even during rest, implying that describing it in a “static” manner might be too simplistic [6,7]. Therefore, the “dynamic functional connectivity (dFC)” has become a new topic in recent years to capture the fluctuations of brain connectivity [8,9,10]. With regard to MDD, some previous studies have provided initial evidence that MDD is associated with a decreased temporal stability (increased temporal variability) of dFC within multiple brain regions such as the medial prefrontal cortex [11,12] and posterior cingulate cortex [13] within the default mode network (DMN), as well as the pallidum [14]. As for studies on the relationships between dFC and suicidality in MDD, several recent studies have found that the overall topological properties of dynamic connectomic [15], the dynamic degree centrality [16], and the dynamic amplitude of low-frequency fluctuation [17] could differentiate the MDD patients with and without suicidal ideation, implying the possible association between the abnormal fluctuations of brain dFC and suicidality in MDD. In a recent study, it was further found that compared with MDD patients without suicidal ideation, those patients with suicidal ideation showed increased dFC variability from the habenula to the superior temporal gyrus and precuneus [18]. Despite the accumulating findings, however, most of the above-mentioned studies are limited by that they only focused on dFC at the level of voxels or within specific regions of interests (ROIs). Recent studies have proved that abnormal dFC patterns in psychiatric disorders are not constrained in a circumscribed area but usually associated with the entire large-scale brain systems [19,20]. Nevertheless, how the changes in dFC within specific large-scale brain networks would be related to the suicidal ideation or behavior in MDD remains largely unknown to our knowledge.

In this study, we aimed to explore the possible relationships between temporal stabilities of large-scale brain dFCs and suicidality in MDD using the dynamic network model and a validated metric, the temporal correlation coefficient [12,21,22,23]. We firstly investigated the alterations in temporal correlation coefficients in MDD patients at the levels of whole brain and nine well-established key networks including the sensorimotor, visual, auditory, default-mode, frontoparietal, cingulo-opercular, salience, subcortical, and attention networks [24,25], respectively. After that, we further investigated their relationships with the level of suicidality in MDD patients. We hypothesized that (1) the previously reported findings in MDD, such as a decreased temporal stability of dFC within the DMN [11,12], would be replicated in the present sample; (2) the temporal stability of dFC within at least one network would be related with the level of suicidality in MDD.

## 2. Materials and Methods

### 2.1. Participants and Measures of Suicidality

The analyzed sample consisted of a total of 52 patients with MDD and 21 age-, sex-matched healthy controls (HCs), who were recruited from the Second Xiangya Hospital of Central South University, Changsha, China. All participants included were right-handed, Han Chinese adults with at least 6 years of education. All patients met the Diagnostic and Statistical Manual of Mental Disorders-IV (DSM-IV) criteria for MDD and had a 17-item Hamilton Rating Scale for Depression (HAMD) score > 7. All participants had no history of any substance abuse, any other neurological disorder, any contraindication to fMRI scanning or any history of electroconvulsive therapy. The HCs additionally met the following inclusion criteria: had no personal or family (in first-degree relatives) history of any mental illness as evaluated by the Structured Clinical Interview for DSM-IV (SCID). Note that the initial sample consisted of 58 MDD patients and 22 HCs; 6 patients and 1 healthy subject were excluded because of excessive head motion (see later in Section 2.2). The study was approved by the Ethics Committee of the Second Xiangya Hospital of Central South University, and written informed consent was obtained from all participants.

For all MDD patients, the level of current suicidality was assessed using the score of the HAMD item 3 (suicidality item), which ranges from 0 to 4. Using a single suicide item from depression scales to assess suicidality in psychiatric disorders has been proved to be valid [26,27,28].

### 2.2. Data Acquisition and Preprocessing

Resting-state fMRI and T1-weighted structural images were acquired on a 3.0 T MRI scanner (Philips Achieva XT). The fMRI images were obtained by gradient echo-planar imaging sequence (repetition time/echo time = 2000/30 ms; slice number = 36; thickness/gap = 4.0/0 mm; field of view = 240 × 240 mm^2^; acquisition matrix = 64 × 64; flip angle = 90°; number of time points = 250), and the T1-weighted images were obtained by three-dimensional fast spoiled gradient recalled sequence (repetition time/echo time = 7.5/3.7 ms; slice number = 180; thickness/gap = 1.0/0 mm; field of view = 240 × 240 mm^2^; acquisition matrix 256 × 200; flip angle = 8°). 

Data preprocessing was performed using the standard pipeline provided by the DPARSF software [29,30]. Briefly, it included removing the first 10 volumes, slice-timing, head motion realignment, brain tissue segmentation, spatial normalization, temporal filtering (0.01–0.10 Hz), and regressing out the signals from white matter, cerebrospinal fluid, and whole brain as well as the Friston-24 head motion parameters [31,32]. All images have been manually checked by trained researchers to ensure good quality. Moreover, 6 patients and 1 healthy subject with excessive head motion were excluded from the analysis, as determined by a mean framewise-displacement (FD) [33] > 0.2 mm.

### 2.3. Dynamic Brain Network Model

The fluctuation of brain dFC patterns was estimated based on the commonly used multilayer dynamic network model [8,9,10]. The nodes in brain network were defined by the Automated Anatomical Labeling (AAL) atlas [34], which was validated in previous fMRI studies [35,36,37]. The names of each of the 90 nodes were listed in the Appendix A (Table A1).

The dynamic networks were constructed as summarized in Figure 1. First, the mean time series were extracted from each node by averaging the signals of all voxels within that node. The widely used sliding-window approach was then applied with a window length of 100 s and an incremental step of 6 s as recommended [12,38,39], dividing the time series into 64 time windows. Within each window, the whole-brain connectivity matrices were calculated using pairwise Pearson correlations. The connectivity matrices were then thresholded with a wide range of densities ranging from 0.01 to 0.50 with an increment interval of 0.01 [40]. For each density, only the connections that survived the given threshold were reserved and assigned a value of 1, and those that did not survive were assigned a value of 0. As a result, a dynamic network G = (G_t_)_t = 1, 2, 3, …, 64_, where G_t_ is the binary subgraph representing brain dFC within the tth time window, was acquired at each density for each subject. The temporal correlation coefficient was then computed at each density separately.

### 2.4. Temporal Correlation Coefficient

Temporal correlation coefficient measures the stability of a dynamic network by the average overlap of all the connections between any two successive time windows as follows: firstly, let *a_ij_* (*t*) = 1 if node *i* and node *j* are connected within the *t*th time window, and *a_ij_* (*t*) = 0 if they are not. The nodal temporal correlation coefficient of the node *i* (*C_i_*) is then defined as
(1)$$C_{i}=\frac{1}{T-1}\sum_{t=1}^{T-1}\frac{\sum_{j}a_{ij}\left(t\right)a_{ij}\left(t+1\right)}{\sqrt{\left[\sum_{j}a_{ij}\left(t\right)\right]\left[\sum_{j}a_{ij}\left(t+1\right)\right]}}$$
where *T* is the number of time windows and N is the number of nodes in the network [12,21,22,41]. The *C_i_* ranges from 0 to 1, and a higher value indicates a higher stability (lower variability) of node *i*. The *C_i_* was computed and averaged across all densities (0.01 to 0.50) to ensure that the results would not be biased by a single threshold [12]. The temporal correlation coefficient of the whole brain is computed by averaging the *C_i_* of all 90 nodes. To further investigate which brain systems were particularly affected, we assigned all nodes to the nine key networks as defined in previous studies [24,25,42] (see details in the Appendix A). The temporal correlation coefficient of each network was computed by averaging the *C_i_* across all nodes of that network [22]. All computations were performed using two publicly available MATLAB toolboxes (https://sites.google.com/site/bctnet and https://github.com/asizemore/Dynamic-Graph-Metrics, accessed on 1 June 2022) [21,43]. The results were partly visualized using the BrainNet Viewer software [44].

### 2.5. Statistics

The demographic characteristics and mean FD were compared between the groups using the two-sample *t*-test or Chi-square test. Temporal correlation coefficients of the whole brain and each of the nine networks were compared between the groups of MDD patients and HCs using analysis of covariance (ANCOVA) covarying for age, sex, and education. The temporal correlation coefficient of whole brain or any network which showed significant between-group differences were further correlated with the total scores of HAMD and the scores of HAMD item 3 using Spearman’s rank correlation coefficient, respectively. For multiple network-level comparisons and multiple correlation tests, the results were corrected by false discovery rate (FDR) and considered significant when the corrected *p* < 0.05.

### 2.6. Post-Hoc Analyses on Clinical Variables

Several post-hoc analyses were performed to explore the potential influence of disease chronicity and medications. First, the temporal correlation coefficients of whole brain or any network with significant between-group differences were correlated with the MDD patients’ illness duration. Second, the temporal correlation coefficients with significant between-group differences were compared between the drug-naïve and medicated MDD patients using the ANCOVA covarying for age, sex, and education. Third, the temporal correlation coefficients with significant between-group differences were correlated with the MDD patients’ daily doses of antidepressant as assessed by fluoxetine equivalents [45]. Significance was set at *p* < 0.05.

### 2.7. Validation Analyses

A number of supplementary analyses were further performed to validate the results. First, all correlation analyses on the temporal correlation coefficients were repeated with patients’ daily doses of antidepressant (fluoxetine equivalents) as an additional covariate (using the partial Spearman’s correlations). Second, the relationships between temporal correlation coefficients and HAMD item 3 scores were investigated using an ordinal logistic regression model [46], to see whether the observed significant relationships would change in different statistical models.

## 3. Results

### 3.1. Demographic Characteristics and Head Motion

As shown in Table 1, there were no significant between-group differences in age and sex (all *p* > 0.05). However, the groups of MDD showed a significantly lower education level (*t* = −3.571, *p* = 0.001). There was no significant between-group difference in head motion as measured by mean FD (*t* = 1.288, *p* = 0.202).

### 3.2. MDD-Related Alterations

As shown in Figure 2, the MDD patients showed a significantly decreased temporal correlation coefficient of the whole brain than HCs (*F* = 8.014, *p* = 0.006); moreover, the MDD patients showed a significantly decreased temporal correlation coefficient of the DMN (*F* = 7.526, FDR-corrected *p* = 0.035) and subcortical network (*F* = 9.885, FDR-corrected *p* = 0.022). The components of the DMN and subcortical network, which showed significant between-group differences, were also presented in Figure 3. No significant between-group differences were found in other networks (all *p* > 0.05).

### 3.3. Correlations

In MDD patients, the scores of HAMD item 3 were found to significantly negatively correlated with the temporal correlation coefficient of the DMN (Spearman’s rho = −0.371, FDR-corrected *p* = 0.021, Figure 4) but not with the temporal correlation coefficient of whole brain or any other network (all *p* > 0.05). No significant correlations were found between the total scores of HAMD and the temporal correlation coefficient of whole brain or any network, either (all *p* > 0.05).

### 3.4. Post-Hoc Analyses on Clinical Variables

No significant correlations were found between the MDD patients’ illness duration and temporal correlation coefficients of the whole brain or any network (all *p* > 0.05). No significant differences were found in temporal correlation coefficients between the drug-naïve and medicated MDD patients (all *p* > 0.05). Additionally, no significant correlations were found between the temporal correlation coefficients and MDD patients’ daily doses of antidepressant assessed by fluoxetine equivalents (all *p* > 0.05).

### 3.5. Validation Analyses

The negative correlation between the temporal correlation coefficient of DMN and scores of HAMD item 3 in MDD patients remained significant when including the patients’ daily doses of antidepressant as an additional covariate (Spearman’s rho = −0.370, *p* = 0.008). Therefore, such a relationship is unlikely to be mainly driven by effects of medication treatment. Furthermore, when using an ordinal logistic regression model, it was still found that the temporal correlation coefficient of DMN is significantly negatively associated with the scores of HAMD item 3 (*z* = −2.494, *p* = 0.013).

## 4. Discussion

The present study, to the best of our knowledge, investigated the alterations in temporal stability of brain dFC and their associations with suicidality in MDD using the dynamic network model and temporal correlation coefficient for the first time. Our major findings were: (1) compared with HCs, the MDD patients exhibited a significantly decreased temporal correlation coefficient of the whole brain, and significantly decreased temporal correlation coefficients of the DMN and subcortical network; (2) the temporal correlation coefficient of the DMN was significantly negatively related with the suicidality in MDD patients. Our findings may expand our understanding of the neurophysiologic mechanisms of the suicidality in MDD.

The decreased temporal correlation coefficient indicates a lower tendency for the brain dFC patterns to be maintained over time or in other words, a decreased temporal stability (increased temporal variability) of dFC [12,21]. Compared with HCs, the MDD patients showed a significantly decreased temporal correlation coefficient of the whole brain, suggesting an excessive fluctuation of FC at the global level. Furthermore, significant local alterations were found in the DMN and subcortical network, which may indicate the prominent excessive fluctuations of these two brain systems (Figure 2). The results were compatible with the prior work which reported an increased variability of FC within the DMN [12,13] and subcortical structures [14] in MDD.

We found that the temporal correlation coefficient of the DMN were significantly negatively related to the scores of the HAMD suicidality item in MDD patients (Figure 4). The result suggests that the excessive fluctuations of dFC within the DMN may underly the psychopathology of suicidality in MDD. The DMN is a brain network which are more active during rest but suppressed during cognitive activities including several distributed nodes such as the precuneus, anterior cingulate cortex, posterior cingulate cortex, and medial prefrontal cortex (Figure 3) [47,48,49]. In MDD patients, static FC patterns within the DMN are often found to be changed [31,49,50] and such changes have been reported to be related to their suicidal thoughts or attempts [3,51,52] in traditional static fMRI studies. The associations between the changes in dFC within the DMN and suicidality in MDD, however, are relatively limited. The DMN were suggested to mediate one’s self-referential and internally directed processing [49,53]. In healthy subjects, the increased variability of dFC within the DMN has been proved to be associated with increased frequencies of spontaneous, internally oriented thoughts such as mind-wandering during the resting state [54,55]. In MDD patients, the increased variability of dFC within the DMN has been also reported to be related with their frequencies of rumination [11,13], a characteristic form of spontaneous, negative, and internally oriented thought defined as “repetitively and passively focusing on symptoms of distress” [56,57]. Therefore, we assume that the excessive variability of the DMN connections may be an indicative of repetitive abnormal activations of the DMN, which lead to excessive negative, internally focused thoughts such as rumination in MDD patients. Such alterations may make the MDD patients more focused on their negative life events and more vulnerable to thinking about suicide, which lead to higher suicidality [58,59]. Our findings, therefore, may extend the finding of the association between abnormal DMN connectivity and suicidality in MDD [3,51,52] into the domain of dynamic brain FC.

In the MDD patients, the decreased temporal stability (increased variability) of dFC as indicated by a significantly decreased temporal correlation coefficient was also observed within the subcortical network including the thalamus, putamen, pallidum and caudate (Figure 4). This result is in line with a recent study which reported an increased variability of dFC in the pallidum in MDD and suggested that it may imply the impaired reward processing [14]. The temporal correlation coefficient of the subcortical network, however, was not significantly correlated with the suicidality in MDD patients. No significant correlation was found between the temporal correlation coefficient of the whole brain and suicidality in MDD, either. Therefore, the excessive variability of dFC within the DMN may be a unique biomarker of the suicidality in MDD.

Our study has several limitations. First, the education level was not matched between the groups of MDD and HCs. To solve this problem, we used the education level as a covariant in the group comparisons to exclude its effect as a potential confounding factor. Second, the suicidality was measured by only the item 3 of HAMD. In the future, a more detailed scale for suicide such as the Columbia Suicide Severity Rating Scale [26] is necessary for more detailed analysis, for example, to distinguish the possible differences be-tween the patients with active and passive suicidal ideation [60]. Third, the sample size in the present study is relatively small and larger samples are needed to confirm our findings. Fourth, the HAMD was not assessed in HCs, and we are unable to confirm whether similar effects would exist in HCs. Fifth, because of the nature of a cross-sectional design, we are unable to establish the causality relationship between the suicidality and brain network stability in MDD patients. Lastly, while a decreased temporal stability of DMN dFC pattern was found in MDD in the present study, it should be noted that reports of an increased stability of DMN dFC in MDD patients also exist [61,62]. It is possible that subtypes of MDD with distinct DMN dFC profiles (hypo- and hyper-stability) could exist [63], which can be investigated in future studies.

## 5. Conclusions

To conclude, using a dynamic brain network model and temporal correlation coefficient, the present study identified a significantly decreased temporal stability (increased variability) of brain dFC at the global level, as well as within the DMN and subcortical network in MDD patients. Moreover, we found that decreased temporal stability of dFC within the DMN was significantly related to higher suicidality in the MDD patients. These findings may have important implications for better understanding and preventing suicide in MDD.

## Figures and Tables

**Figure 1 brainsci-12-01263-f001:**
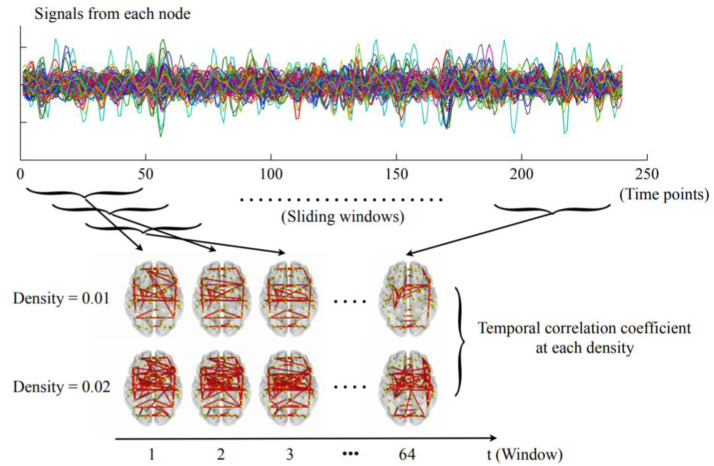
The steps for constructing dynamic networks and calculating temporal correlation coefficients as detailed in the Materials and Methods section (only the densities of 0.01 and 0.02 were presented here for visualization purposes).

**Figure 2 brainsci-12-01263-f002:**
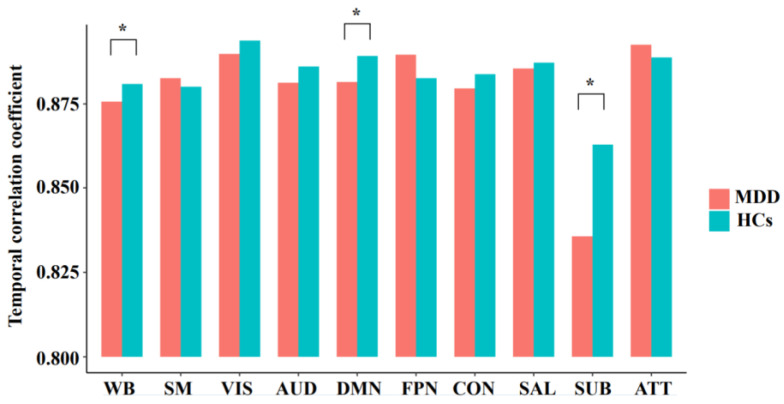
The mean temporal correlation coefficients of the whole brain and each network in the groups of major depressive disorder (MDD) and healthy controls (HCs), after adjusting for age, sex, and education. WB, whole brain; SM, sensorimotor network; VIS, visual network; AUD, auditory network; DMN, default mode network; FPN, frontoparietal network; CON, cingulo-opercular network; SAL, salience network; SUB, subcortical network; ATT, attention network; * significant difference with corrected *p* < 0.05.

**Figure 3 brainsci-12-01263-f003:**
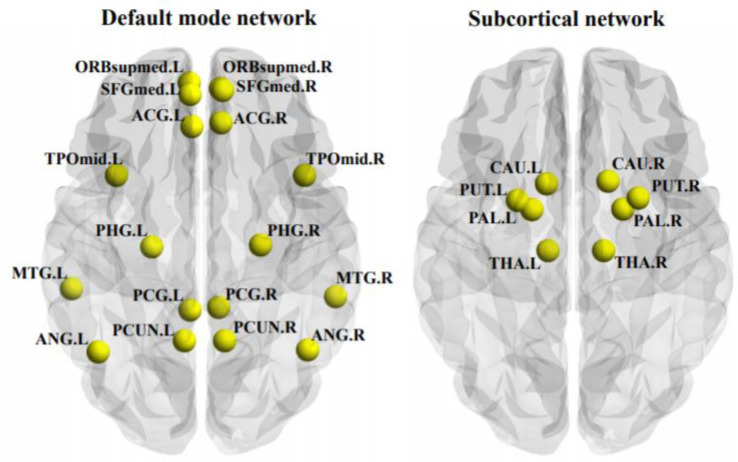
The components of the default mode network and subcortical network, of which the temporal correlation coefficients were significantly decreased in the patients with major depressive disorder. ACG, anterior cingulate cortex; ANG, angular gyrus; CAU, caudate; L, left hemisphere; MTG, middle temporal gyrus; ORBsupmed, medial orbitofrontal cortex; PAL, pallidum; PCUN, precuneus; PHG, parahippocampal gyrus; PUT, putamen; R, right hemisphere; SFGmed, medial superior frontal cortex; SPG, superior parietal gyrus; THA, thalamus; TPOmid, middle temporal pole.

**Figure 4 brainsci-12-01263-f004:**
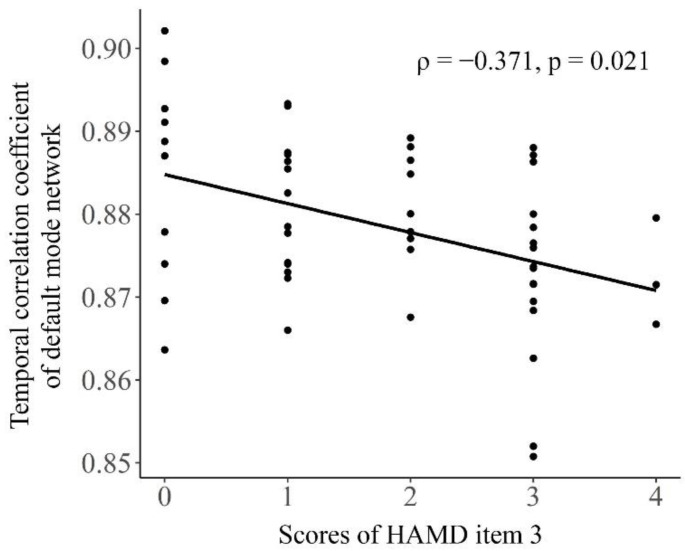
The relationship between the temporal correlation coefficient of default mode network and the scores of HAMD item 3 in the patients with major depressive disorder. The Spearman’s correlation coefficient (ρ) and FDR-corrected *p* value were presented on the figure.

**Table 1 brainsci-12-01263-t001:** Demographic, clinical, and head motion characteristics of the groups. SD, standard deviation; MDD, major depressive disorder; HCs, healthy controls; HAMD, Hamilton Rating Scale for Depression; FD, framewise-displacement.

	MDD (*n* = 52) (Mean ± SD)	HCs (*n* = 21) (Mean ± SD)	Group Comparisons
Age (years)	30.615 ± 9.306	27.000 ± 5.908	*t* = 1.982, *p* = 0.052
Sex (male/female)	28/24	10/11	*χ*^2^ = 0.232, *p* = 0.796
Education (years)	12.115 ± 3.191	14.952 ± 2.747	*t* = −3.571, *p* = 0.001
17-item HAMD scores	20.789 ± 6.366	/	/
HAMD item 3 scores	1.769 ± 1.246	/	/
Illness duration (months)	42.919 ± 62.291	/	/
Drug-naïve/medicated	9/43	/	/
Fluoxetine equivalents (mg/d)	25.360 ± 20.024		
Mean FD	0.086 ± 0.038	0.074 ± 0.030	*t* = 1.288, *p* = 0.202

## Data Availability

The data presented in this study are available on request from the corresponding author.

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
