# Peer review of "Temporal Stability of Dynamic Default Mode Network Connectivity Negatively Correlates with Suicidality in Major Depressive Disorder"

_brainsci, 2022, doi:10.3390/brainsci12091263_

Round 1

Reviewer 1 Report

This study aimed to explore the relationships between temporal stabilities of large-scale brain dynamic functional connectivity and suicidality in 52 major depressive disorder and 21 healthy controls using the dynamic network model and a validated metric, the temporal correlation coefficient. The authors found that a significantly decreased temporal stability of brain dFC at the global level, as well as within the DMN and subcortical  network in MDD patients. And they found that decreased temporal stability of dFC within the DMN was significantly related to higher suicidality in the MDD patients.

I have some comments:

-       More details are needed on how healthy controls were assessed and their family history investigated

-       There is no need of stating the sample size is of 58 patients and 22 healthy controls. Since 6 patients and 1 control were excluded from the analysis, the sample size is 52 and 21.

-       I wonder how robust is this methodology when small sample size are considered (for instance drug naïve are just 9). In addition the absence of differences does not imply that drugs might not impact the validity of the findings, also because quite likely patients who had higher scores at item 3 were all under treatment. Thus the effect of treatment should be a covariate in the analysis of the relationship between temporal stability and suicidality.

-       The item 3 ranges from 0 to 4 (not to 3). In addition, it is a likert scake and using correlations coefficient might not be appropriate, an ordinal logistic regression might work better

Author Response

Dear Reviewer 1,

Thank you very much for your constructive critique to improve our manuscript. We have made every effort to address the issues raised and to respond to all comments. Please find the detailed, point-by-point responses below. We hope that our revisions would meet your expectations.

(1) “More details are needed on how healthy controls were assessed and their family history investigated.”

Response: In accordance with your comment, we have supplemented the following details in “2.5. Participants and Measures of Suicidality” subsection: “All participants included were right-handed, Han Chinese adults with at least 6 years of education. All participants had no history of any substance abuse, any other neurological disorder, any contraindication to fMRI scanning or any history of electroconvulsive therapy… The HCs additionally met the following inclusion criteria: had no personal or family (in first-degree relatives) history of any mental illness as evaluated by the Structured Clinical Interview for DSM-IV (SCID).” (Lines 78–88)

Moreover, please note that the HAMD was not assessed in HCs in the present study, which might be considered as a limitation. We have supplemented the following sentence in the Discussion section: “Fourth, the HAMD was not assessed in HCs, and we are unable to confirm whether similar effects would exist in HCs.” (Lines 324–325)

(2) “There is no need of stating the sample size is of 58 patients and 22 healthy controls. Since 6 patients and 1 control were excluded from the analysis, the sample size is 52 and 21.”

Response: In accordance with your comment, we have revised the related descriptions in the manuscript. The revised descriptions are as follows:

“The analyzed sample consisted of a total of 52 patients with MDD and 21 age-, sex- matched healthy controls (HCs), who were recruited from the Second Xiang-ya Hospital of Central South University, Changsha, China… Note that the initial sample consisted of 58 MDD patients and 22 HCs; 6 patients and 1 healthy subject were excluded because of excessive head motion.” (Lines 76–91)

(3) “I wonder how robust is this methodology when small sample size are considered (for instance drug naïve are just 9). In addition the absence of differences does not imply that drugs might not impact the validity of the findings, also because quite likely patients who had higher scores at item 3 were all under treatment. Thus the effect of treatment should be a covariate in the analysis of the relationship between temporal stability and suicidality.”

Response: Thank you very much for your valuable comments. Although the sample size is relatively small, we believe that our results are reliable since the methodology strictly followed the previously published studies, and the results were strictly corrected for multiple tests (e.g., multiple network-level comparisons). However, we agree that it would be meaningful to confirm our results in a larger sample in future studies for more robust conclusions. The following sentence has been included in the manuscript: “…Third, the sample size in the present study is relatively small and larger samples are needed to confirm our findings.” (lines 322-324)

We agree that it is necessary to validate our results by excluding the possible confounding effects of medication treatment. Therefore, we have added some additional validation analyses in the revised manuscript, to prove that the results were not mainly driven by medication treatment, as follows:

In the Methods section: “…Third, the temporal correlation coefficients with significant between-group differences were correlated with the MDD patients’ daily doses of antidepressant as assessed by fluoxetine equivalents [45].” (Lines 191–193); and “…all correlation analyses on the temporal correlation coefficients were repeated with patients’ daily doses of antidepressant (fluoxetine equivalents) as an additional covariate” (Lines 195-198). In the Results section: “…no significant correlations were found between the temporal correlation coefficients and MDD patients’ daily doses of antidepressant assessed by fluoxetine equivalents (all p > 0.05)” (Lines 250-252); and “The negative correlation between the temporal correlation coefficient of DMN and scores of HAMD item 3 in MDD patients remained significant when including the patients’ daily doses of antidepressant as an additional covariate (Spearman's rho = -0.370, p = 0.008). Therefore, such a relationship is unlikely to be mainly driven by effects of medication treatment.” (Lines 254-258).

(4) “The item 3 ranges from 0 to 4 (not to 3). In addition, it is a likert scake and using correlations coefficient might not be appropriate, an ordinal logistic regression might work better.”

Response: Thank you very much for your valuable comments. We have revised “ranges from 0 to 3” to “ranges from 0 to 4” in Line 95.

Please note that we used Spearman’s rank correlations rather than Pearson correlations in the present study. The Spearman’s rank correlations should be appropriate for Likert scales. However, we agree that it would be better to confirm the results with an ordinal logistic regression model. We have supplemented an additional validation analysis using the ordinal logistic regression model as follows:

“…the relationships between temporal correlation coefficients and HAMD item 3 scores were investigated using an ordinal logistic regression model [46], to see whether the observed significant relationships would change in different statistical models” (Lines 198-201); and “Furthermore, when using an ordinal logistic regression model, it was still found that the temporal correlation coefficient of DMN is significantly negatively associated with the scores of HAMD item 3 (z = -2.494, p = 0.013)” (Lines 258-260).

Reviewer 2 Report

The paper is interesting and can be published.  Some limitation can be the association of results from a Mri and psychological scales. These association can be a little "forced" however the paper is interesting and deserve to be published.

Author Response

Dear Reviewer 2,

Thank you very much for your constructive critique to our manuscript. We agree that it would be an important limitation that althouth significant correlations were observed, we are still unable to establish the causality relationship between the suicidality and brain network stability in MDD patients. Therefore, we have supplemented the following discussions in the revised manuscript: "because of the nature of a cross-sectional design, we are unable to establish the causali-ty relationship between the suicidality and brain network stability in MDD patients" (Lines 325-327). Besides, some revisions have been made based on other Reviewers' comments and suggestions. We hope that our revisions would meet your expectations.

Reviewer 3 Report

The manuscript entitled: “Temporal Stability of Dynamic Default Mode Network Connectivity Negatively Correlates with Suicidality in Major Depressive Disorder” by Xuan Ouyang et al. (Manuscript ID: brainsci-1903175) describes the changes in dynamic functional connectivity (dFC) (using the resting-state functional magnetic resonance imaging /fMRI/  between specific brain regions and the level of the suicidality (using scores on the item 3 from the 17-item Hamilton Rating Scale for Depression/HAMD/) in 52 patients with major depressive disorder (MDD) and 21 healthy controls (HCs).

This is a well written and interesting MS and I have no additional comments for the authors.

Author Response

Dear Reviewer 3,

Thank you very much for your positive comments. Although you didn't give additional suggestions, some revisions have been made based on other Reviewers' comments and suggestions. We hope that our revisions would meet your expectations.

Round 2

Reviewer 1 Report

No further comments.